# 🍯 DEL: Context-Aware *D*ynamic *E*xit *L*ayer for Efficient Self-Speculative Decoding

**Hossein Entezari Zarch**[*], **Lei Gao**[*], **Chaoyi Jiang** , **Murali Annavaram**
University of Southern California
{entezari, leig, chaoyij, annavara}@usc.edu

## Abstract

Speculative Decoding (SD) is a widely used approach to accelerate the inference of large language models (LLMs) without reducing generation quality. It operates by first using a compact model to draft multiple tokens efficiently, followed by parallel verification using the target LLM. This approach leads to faster inference compared to auto-regressive decoding. While there are multiple approaches to create a draft model, one promising approach is to use early-exit methods. These methods draft candidate tokens by using a subset of layers of the primary model and applying the remaining layers for verification, allowing a single model to handle both drafting and verification. While this technique reduces memory usage and computational cost, its performance relies on the choice of the exit layer for drafting and the number of tokens drafted (speculation length) in each SD round. Prior works use hyperparameter exploration to statically select these values. However, our evaluations show that these hyperparameter values are task-specific, and even within a task they are dependent on the current sequence context. We introduce DEL (Dynamic Exit Layer), a plug-and-play method that adaptively selects the exit layer and speculation length during inference. DEL dynamically tracks the token acceptance rate if the tokens are drafted at each layer of an LLM and uses that knowledge to heuristically select the optimal exit layer and speculation length. Our experiments across a broad range of models and downstream tasks show that DEL achieves overall speedups of $2.16\times\sim2.62\times$ over vanilla auto-regressive decoding and improves upon state-of-the-art SD methods, which peak at $2.43\times$, by up to $0.19\times$. The code is available at https://github.com/hoenza/DEL.

## 1 Introduction

Large language models (LLMs) such as Llama 2 (Touvron et al., 2023), Llama 3 (Dubey et al., 2024), GPT-4 (OpenAI et al., 2024), and Claude (Bai et al., 2022) have demonstrated strong performance across diverse tasks, driving adoption in search engines (Microsoft, 2023; Reid, 2023), chatbots (OpenAI, 2022), and virtual assistants (Wu et al., 2023a;b). However, LLM token generation remains slow due to its auto-regressive nature, where each token is generated sequentially based on all previous tokens. This sequential dependency results in low generation speed, as every new token requires access to the full set of model parameters. The resulting I/O bottleneck limits hardware utilization during inference, especially in small-batch or low-latency scenarios, such as edge deployments or real-time applications.

Speculative decoding (SD) (Leviathan et al., 2023; Chen et al., 2023) addresses this bottleneck by using a smaller model to draft a sequence of tokens auto-regressively, which are then verified in parallel by the target model. This reduces latency while preserving the output distribution of the main model. The performance of SD depends on two key factors: the speed and accuracy of the draft model, and the number of tokens drafted per SD round (speculation length $\gamma$). Larger draft models increase acceptance rates but reduce draft speed.

---

[*]Equal contribution.

A larger $\gamma$ increases the number of proposed tokens per round, which may lead to more accepted tokens, but it also increases the likelihood of rejection, potentially reducing overall efficiency. Early versions of SD relied on separate draft models with fixed $\gamma$, tuned offline. Later work explored adaptive $\gamma$ policies (Zhang et al., 2024).

LayerSkip (Elhoushi et al., 2024) is a recent self-speculative decoding method that improves efficiency by removing the need for a separate draft model. Instead, it reuses the first $\mathcal{E}$ layers of the main model to draft tokens and the remaining layers for verification, reducing both memory and compute overhead. Prior works have relied on static configuration of $\mathcal{E}$ and $\gamma$, selected via offline grid search. This introduces two key limitations. First, the optimal $\mathcal{E}$ and $\gamma$ vary significantly across tasks; configurations tuned for one task often underperform on others. For example, using a language modeling–optimized configuration on a summarization task reduces speedup from $2.65\times$ to $1.50\times$. Second, even within a single task, optimal configurations can shift during generation. In code generation, for instance, a setting that performs well for early tokens may degrade performance later in the sequence. These findings indicate that fixed configurations are suboptimal and motivate the need for a dynamic approach that adapts $\mathcal{E}$ and $\gamma$ throughout the generation process.

Coincidentally, we observe that LayerSkip is particularly well-suited for dynamic selection of $\mathcal{E}$ and $\gamma$ with minimal overhead. First, since any prefix of the model layers can serve as a draft model, LayerSkip naturally exposes a spectrum of sub-models defined by the choice of exit layer $\mathcal{E}$, with no overhead when switching between them during generation. Second, for each draft token, LayerSkip computes hidden states of the first $\mathcal{E}$ model layers during the drafting stage, and then computes the remaining layers hidden states during verification in each SD round. Importantly, these intermediate hidden states computed can be reused to estimate the token acceptance rate under different choices of $\mathcal{E}$. Together, these two properties enable real-time adaptation of $\mathcal{E}$ and $\gamma$ during inference, guided by layer-wise acceptance rate estimates. To fully exploit this flexibility, however, one needs an algorithmic approach to adaptively select appropriate values for $\mathcal{E}$ and $\gamma$ at each SD round.

To address this challenge, we introduce DEL (Dynamic Exit Layer), an algorithmic approach that is implemented as a plug-and-play module for LayerSkip that adaptively selects $\mathcal{E}$ and $\gamma$ at each round of speculative decoding. DEL tracks the acceptance rate of draft tokens for each layer using cached hidden states, and uses this information to estimate a metric called Token-per-Layer (TPL), which reflects decoding efficiency. By selecting the configuration that maximizes TPL, DEL improves speed without sacrificing output quality. DEL also adjusts $\gamma$ dynamically within each round using a context-aware confidence threshold to decide when to stop drafting.

In summary, our key contributions are:

- We conducted an empirical study on the impact of $\mathcal{E}$ and $\gamma$ in LayerSkip, revealing that the optimal settings of these values are model- and input-dependent, motivating the need for dynamic selection of these parameters during inference.

- Based on these observations, we propose DEL, a plug-and-play module for LayerSkip that adaptively selects $\mathcal{E}$ and $\gamma$ by evaluating cached hidden states and applying a dynamic, context-aware thresholding mechanism.

- Through extensive evaluations across tasks and model sizes, we show that DEL achieves up to $2.62\times$ speedup over vanilla decoding and outperforms dynamic LayerSkip baselines, which peak at $2.43\times$, while maintaining output quality and incurring negligible overhead.

## 2 Background

**Auto-regressive decoding.** Given an input sequence $\{x_0, \dots, x_{t-1}\}$, a transformer-based auto-regressive language model $\mathcal{M}_p$ generates the next token $x_t$ from the conditional probability distribution $p(x_t \mid x_{<t})$. Specifically, each transformer decoder layer processes the hidden states $h_t^{\ell-1}$ from the previous layer as follows:

$$h_t^\ell = \text{Transformer}_\ell(h_t^{\ell-1}), \quad \ell \in [1, L],$$

where $h_t^0$ is the embedding representation of $x_{t-1}$. After the $L$-th layer of the model, the predicted token $x_t$ is determined by the probability output from a softmax classifier: $p(x_t) = \text{LM-Head}(h_t^L) = \text{softmax}(W^\top h_t^L)$. We use the notation $p(x_t)$ to denote $p(x_t \mid x_{<t})$.

**Speculative decoding.** The SD algorithm generates tokens in two phases: drafting and verification. During the drafting phase, a small and efficient draft model $\mathcal{M}_q$ is used to sample $\gamma$ candidate token distributions $\{q(x_t), \ldots, q(x_{t+\gamma-1})\}$. In the verification phase, the input sequence concatenated with these $\gamma$ drafted tokens is passed to the target model $\mathcal{M}_p$, which computes the probabilities $\{p(x_t), \ldots, p(x_{t+\gamma})\}$ in parallel. The verification step compares $p(x_{t+i})$ and $q(x_{t+i})$, and accepts the drafted token with probability $\min(1, p(x_{t+i})/q(x_{t+i}))$.

If a token $x_{t+i}$ is rejected before all $\gamma$ tokens are accepted, the remaining draft tokens are discarded, and $x_{t+i}$ is resampled from the residual distribution $\max(0, p(x_{t+i}) - q(x_{t+i}))$. Otherwise, if all drafted tokens are accepted, SD samples one additional token from $p(x_{t+\gamma})$ and appends it to the sequence. Each round of SD thus generates at least one and at most $\gamma + 1$ tokens. Leviathan et al. (2023) prove that the output sequence from SD follows the same distribution as that of the target model. In the special case of greedy decoding, the verification phase simply compares the top-1 prediction from $p(x_{t+i})$ and $q(x_{t+i})$. A key advantage of SD is that the target model verifies all draft tokens in parallel, offering higher efficiency than generating tokens sequentially.

**Early-exit self-speculative decoding.** LayerSkip (Elhoushi et al., 2024) removes the need for a separate draft model $\mathcal{M}_q$ by leveraging a subset of the full model as the draft model. During decoding, only the first $\mathcal{E}$ layers are executed, and the output is passed directly to the LM-Head to generate a draft token. Once $\gamma$ tokens are drafted auto-regressively, they are passed through all $L$ layers of the model for verification. LayerSkip's verification phase reuses the KV cache from the draft phase, resulting in lower memory usage and reduced computation compared to other SD approaches that rely on independent draft models. Nevertheless, the performance of LayerSkip is sensitive to the choice of $\mathcal{E}$ and $\gamma$.

## 3 Motivating Observation

**Exit layer $\mathcal{E}$ and speculation length $\gamma$ are task-dependent.** We begin by investigating whether a single configuration of exit layer $\mathcal{E}$ and speculation length $\gamma$ can generalize across tasks. To this end, we perform a grid search to identify the optimal $(\mathcal{E}, \gamma)$ for each task, selecting the configuration that maximizes decoding speed (measured in tokens per second). We then apply each task-specific configuration to all other tasks to assess the generalizability of these settings. Figure 1(a) shows that applying a configuration tuned for one task to another often degrades performance. For instance, using the language modeling configuration for summarization reduces speedup from $2.65\times$ to $1.50\times$, while applying the summarization configuration to language modeling drops speedup from $2.01\times$ to $1.54\times$.

**Even within a single task, optimal settings vary by prompt and generation stage.** To investigate whether optimal $(\mathcal{E}, \gamma)$ settings remain stable across prompts within a task, or even across the course of generating output from a single prompt, we conduct a finer-grained analysis. We randomly sample two prompts from the coding task and divide the generation process into fixed-length segments of 64 tokens. For each segment, we run a separate grid search to determine the optimal configuration for that specific generation window. Figure 1(b) visualizes the results as heatmaps, where the first row of three figures correspond to the first prompt, and the second row corresponds to the second prompt. The columns within each row correspond to the heat map generated for a particular segment. The heatmaps reveal that the best-performing configuration shifts not only between prompts but also across different stages of token generation within the same prompt. Notably, the globally optimal configuration for the coding task, $(\mathcal{E}=7, \gamma=6)$, often fails to achieve the best performance at finer granularity, as several segments exhibit higher decoding speed with alternative settings.

These findings highlight a critical insight: the optimal exit layer $\mathcal{E}$ and speculation length $\gamma$ are not solely functions of the underlying task, and they also depend heavily on the specific prompt and the dynamic context during generation. In practice, the optimal configuration

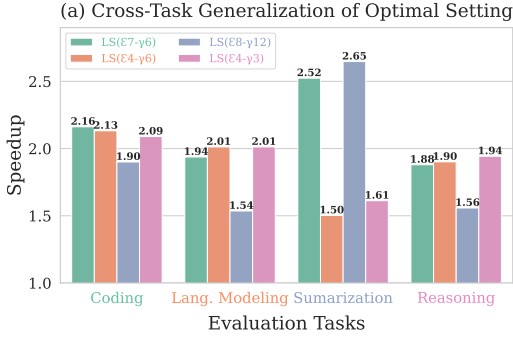 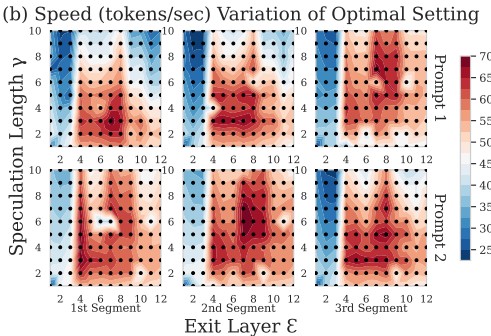

Figure 1: (a) Optimal $(\mathcal{E}, \gamma)$ configurations of LayerSkip are task-specific; there is no universally best setting. The legend shows configurations tuned for each task, with colors matching the corresponding task label on the x-axis. Bars show the speedup achieved when applying each configuration to each task. Tasks perform best with their own configuration, while cross-task settings degrade performance. (b) Optimal $(\mathcal{E}, \gamma)$ configurations vary across prompts and even within segments of a single prompt. Heatmaps show decoding speed over $(\mathcal{E}, \gamma)$ grids. Shifting optimal regions motivate dynamic, context-aware adaptation.

often shifts throughout the generation process, even within a single prompt. This motivates our core contribution: a dynamic, context-aware approach that adaptively selects $\mathcal{E}$ and $\gamma$ at fine granularity for each stage of generation, in order to fully realize the potential efficiency of self-speculative decoding.

# 4 DEL

As shown in the prior section the token generation speed is a function of exit layer $\mathcal{E}$ and speculation length $\gamma$. A draft model with a larger value of $\mathcal{E}$ may increase the likelihood that a drafted token is accepted, but it also introduces higher latency for drafting each token. Similarly, a larger value of $\gamma$ may reduce the total number of SD rounds but can also increase the number of redundant draft generations. Hence, one needs an appropriate token generation cost model to evaluate the trade-off introduced by $\mathcal{E}$ and $\gamma$.

## 4.1 Defining TPL Metric

We introduce *Token-per-Layer (TPL)*, a metric that estimates the expected number of tokens generated per loaded layer in each SD round for the LayerSkip approach. TPL is defined as:

$$\text{TPL}(\ell, d) = \frac{\text{\# generated tokens}}{\text{\# loaded layers}} = \frac{1 - \alpha_\ell^{d+1}}{(1 - \alpha_\ell)(d\ell + L)} \tag{1}$$

where $\alpha_\ell$ is the expected acceptance rate when using an exit layer $\ell$ to draft the tokens. As described in prior work (Leviathan et al., 2023) if the probability that a token proposed by the draft model with $\ell$ layers is accepted is $\alpha_\ell$ then the expected number of newly generated tokens when using speculation length of $d$ is given by $\frac{1-\alpha_\ell^{d+1}}{1-\alpha_\ell}$. Note that this equation is a simplification of $\sum_{i=0}^{d}(\alpha_\ell)^i$. We take this expected token generation length and divide that by the cost to generate these $d$ tokens to create the TPL metric.

**Cost estimation.** While there are various approaches to measure the cost of generating tokens, we rely on an easily measurable approximation. In particular, we exploit a known observation that LLM token generation is heavily memory-bounded during inference, and the latency of generating a token is proportional to the number of transformer layers loaded into the GPU (Xia et al., 2024). Loading a model layer and forwarding multiple tokens for verification takes roughly the same amount of time as loading a model layer and forwarding a single token for drafting. By exploiting this observation, we estimate the cost of drafting

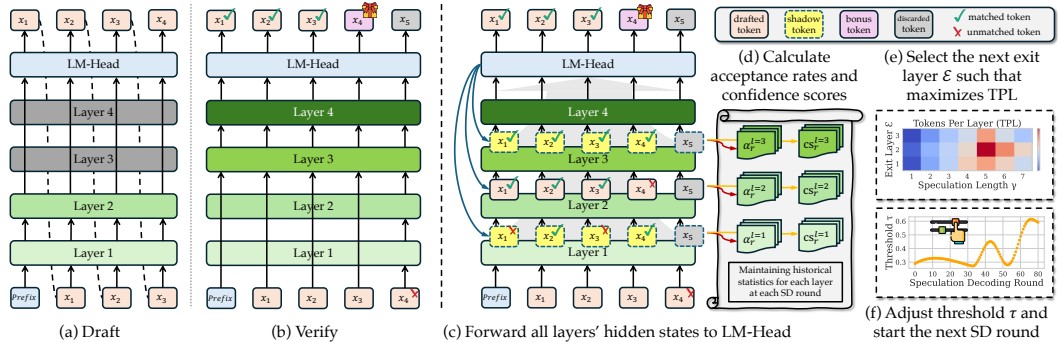

Figure 2: DEL Overview. (a–b) In this LayerSkip example, Layer 2 serves as the exit layer, drafting 4 tokens auto-regressively. During verification, 3 tokens are accepted, while $x_4$ is rejected and becomes the bonus token. (c–d) The plug-and-play DEL module reuses the hidden states from all layers, forwarding them through the LM-Head to generate shadow tokens used to estimate layer-wise token acceptance rates and confidence scores without incurring additional forward passes. (e–f) DEL selects the next exit layer $\mathcal{E}$ that maximizes Token-per-Layer (TPL) based on historical acceptance statistics, and updates the dynamic confidence threshold $\tau$ to guide adaptive speculation length $\gamma$ in the next SD round.

and verifying $d$ tokens at each SD round with exit layer $\ell$ as $(d\ell + L)$, where the draft model with $\ell$ layers is loaded $d$ times, and the target model with $L$ layers is loaded once. This cost estimation is used to compute the TPL metric discussed above.

## 4.2 Modeling Expected Acceptance Rate

To accurately estimate TPL, as defined in Eq. (1), our methodology requires modeling the expected token acceptance rate $\alpha_\ell$ for each layer. Different layers yield different values of $\alpha_\ell$ and incur different drafting costs, leading to different TPL values. To enable efficient and adaptive estimation of $\alpha_\ell$ in LayerSkip, we make the following key observation. One can directly utilize the hidden states produced in each SD round to efficiently compute the $\alpha_\ell$ for each layer, as we discuss next.

**Shadow tokens generation.** Specifically, during the drafting and verification phases of LayerSkip (Figure 2(a) and (b)), we cache all generated hidden states across all layers, denoted as $\{h_t^\ell, \dots, h_{t+\gamma}^\ell\}$, for $\ell \in [1, L]$. As shown in Figure 2(c), we then forward these hidden states through the LM-head in a single pass and apply greedy decoding to obtain $\{x_t^\ell, \dots, x_{t+\gamma}^\ell\}$, producing $L$ sets of tokens, one set for each layer. Each set represents the candidate tokens (referred to as *shadow tokens*) that would have been generated if the corresponding layer had been used as the draft model. These shadow tokens allow us to estimate the expected acceptance rate $\alpha_\ell$ for each possible exit layer.

**$\alpha_\ell$ formulation.** Suppose layer $E$ is used as the draft model during the drafting phase of SD round $r$. We compare the tokens $\{x_t^E, \dots, x_{t+\gamma}^E\}$, generated by the draft model, with $\{x_t^L, \dots, x_{t+\gamma}^L\}$, generated by the target model. We then identify the index of the first mismatched token: $u_r = \min(\{i \in [0, \gamma] : x_{t+i}^E \neq x_{t+i}^L\})$, if a mismatch exists; otherwise, $u_r = \gamma$. The index $u_r$ indicates the point of divergence between the draft and target generations. Tokens beyond this point are discarded, as the context has diverged from the target model's generation. For each layer $\ell \in [1, L]$, the number of matched shadow tokens within the valid context is computed as $c_r^\ell = \sum_{i=0}^{u_r} \mathbb{I}(x_{t+i}^\ell = x_{t+i}^L)$, where $\mathbb{I}(\cdot)$ denotes the indicator function.

The above procedure provides $c_r^\ell$ and $u_r$ for the current SD round $r$. Our goal is to estimate $\alpha_\ell$ by tracking this information over multiple past rounds. To achieve this, we define a weighted sum function $S(a) = \sum_{i=0}^{r-1} \omega^{r-i-1} a[i]$, where $a \in \mathbb{R}^r$ is the input vector and

$\omega \in [0, 1]$ is a decay control factor. The expected acceptance rate is then defined as:

$$\alpha_\ell = \frac{S(\boldsymbol{c}^\ell)}{S(\boldsymbol{u})}, \quad \ell \in [1, L), \tag{2}$$

where $\boldsymbol{c}^\ell, \boldsymbol{u} \in \mathbb{R}^r$ represents the number of matched shadow tokens and length of the valid context for layer $\ell$ across SD rounds up to $r$, respectively. The expected acceptance rate $\alpha_\ell$ is computed as a weighted ratio of matched tokens to total tokens, with recent rounds weighted more heavily. As a result, $\alpha_\ell$ reflects how well the predictions from a given exit layer align with the target model over time.

### 4.3 Selecting Exit Layer Dynamically

With the estimated $\alpha_\ell$, we evaluate TPL$(\ell, d)$ for all possible values of $\ell \in [1, L)$ and $d \in [0, d_{\max}]$. The optimal setting for the next SD round is selected such that TPL$(\ell{=}\mathcal{E}, d{=}\gamma)$ is maximized. To determine the initial values of $\mathcal{E}$ and $\gamma$ for the first SD round, we begin inference with the pre-filling step, where the target model generates initial hidden states for all layers over the prompt context to evaluate TPL.

While $\mathcal{E}$ and $\gamma$ are selected at the SD round granularity based on contextual information and historical statistics, the speculation length $\gamma$ can still be suboptimal for individual tokens due to variation in token-level predictability within an SD round. To address this, we introduce a dynamic draft-exiting mechanism that allows the draft process to be shortened or extended based on the confidence of individual draft tokens, enabling finer-grained and more accurate control within each SD round.

### 4.4 Dynamic Draft-Exiting

The probability of the top-1 draft token prediction is used as a confidence score, representing the likelihood that the token will be accepted during verification (Du et al., 2024; Xia et al., 2025). In DEL, drafting proceeds for up to $d_{\max}$ steps but may stop earlier if the confidence score falls below a threshold. While prior work uses a fixed threshold, this approach may not generalize well across prompts. Challenging prompts might require higher thresholds, while simpler ones may tolerate lower values, and the optimal threshold can even vary within a single prompt. To address this, we adopt a dynamic threshold that adjusts based on a context-aware update rule.

Let $\{cs_t^\ell, \ldots, cs_{t+\gamma}^\ell\}$ denote the confidence scores for the shadow tokens $\{x_t^\ell, \ldots, x_{t+\gamma}^\ell\}$ at layer $\ell$. During SD round $r$, the sum of confidence scores over matched tokens within the valid context from layer $\ell$ is defined as $tcs_r^\ell = \sum_{i=0}^{u_r} cs_{t+i}^\ell \cdot \mathbb{I}(x_{t+i}^\ell = x_{t+i}^L)$. Similarly, the sum of confidence scores over mismatched tokens from layer $\ell$ is $fcs_r^\ell = \sum_{i=0}^{u_r} cs_{t+i}^\ell \cdot \mathbb{I}(x_{t+i}^\ell \neq x_{t+i}^L)$. We then define the dynamic threshold as follows:

$$\tau_\ell = \frac{1}{2} \left( \frac{S(\boldsymbol{tcs}^\ell)}{S(\boldsymbol{c}^\ell)} + \frac{S(\boldsymbol{fcs}^\ell)}{S(\boldsymbol{u} - \boldsymbol{c}^\ell)} \right), \quad \ell \in [1, L), \tag{3}$$

where $\boldsymbol{tcs}^\ell, \boldsymbol{fcs}^\ell \in \mathbb{R}^r$ denote the historical confidence score sums for matched and mismatched tokens across SD rounds up to $r$, and $\boldsymbol{c}^\ell, \boldsymbol{u} - \boldsymbol{c}^\ell \in \mathbb{R}^r$ represent number of matched and mismatched tokens over the same rounds. Therefore, the dynamic threshold $\tau_\ell$ is computed as the midpoint between the weighted average confidence scores of matched and mismatched tokens, with greater emphasis on recent rounds. We provide the summarization of the DEL method in Appendix A.

## 5 Experiments

### 5.1 Experimental Setup

**Implementation details.** We evaluate DEL on top of LayerSkip (Elhoushi et al., 2024) using LLaMA-2, LLaMA-3, and CodeLLaMA models across a range of tasks: CNN/DailyMail

| Models | Methods | AQuA-RAT (Reasoning) | | CNN/DM (Lang. Mod.) | | CNN/DM (Abs. Sum.) | | XSUM (Abs. Sum.) | | Speed (tokens/s) | Overall Speedup |
|---|---|---|---|---|---|---|---|---|---|---|---|
| | | eTPL | Speedup | eTPL | Speedup | eTPL | Speedup | eTPL | Speedup | | |
| LLaMA -3.2-1B | Vanilla | 0.063 | 1.00× | 0.063 | 1.00× | 0.063 | 1.00× | 0.063 | 1.00× | 64.34 | 1.00× |
| | LS($\mathcal{E}$3-$\gamma$6) | 0.146 | 2.01× | 0.154 | 2.11× | 0.144 | 1.93× | 0.139 | 1.89× | 127.87 | 1.99× |
| | FS($\mathcal{E}$3-$\gamma$6) | 0.158 | 2.16× | 0.184 | 2.48× | 0.159 | 2.13× | 0.159 | 2.18× | 144.21 | 2.24× |
| | DV($\mathcal{E}$3) | 0.122 | 1.72× | 0.138 | 1.93× | 0.132 | 1.83× | 0.099 | 1.43× | 111.14 | 1.73× |
| | DEL | 0.170 | **2.24×** | 0.198 | **2.56×** | 0.173 | **2.25×** | 0.183 | **2.37×** | 151.73 | **2.36×** |
| LLaMA -3-8B | Vanilla | 0.031 | 1.00× | 0.031 | 1.00× | 0.031 | 1.00× | 0.031 | 1.00× | 32.57 | 1.00× |
| | LS($\mathcal{E}$3-$\gamma$6) | 0.084 | 2.17× | 0.087 | 2.29× | 0.089 | 2.30× | 0.077 | 2.08× | 72.00 | 2.21× |
| | FS($\mathcal{E}$3-$\gamma$6) | 0.094 | 2.47× | 0.102 | 2.76× | 0.087 | 2.30× | 0.080 | 2.18× | 79.13 | 2.43× |
| | DV($\mathcal{E}$3) | 0.052 | 1.52× | 0.052 | 1.50× | 0.085 | 2.32× | 0.051 | 1.46× | 55.40 | 1.70× |
| | DEL | 0.098 | **2.53×** | 0.109 | **2.81×** | 0.105 | **2.72×** | 0.092 | **2.40×** | 85.20 | **2.62×** |
| LLaMA -2-7B | Vanilla | 0.031 | 1.00× | 0.031 | 1.00× | 0.031 | 1.00× | 0.031 | 1.00× | 36.08 | 1.00× |
| | LS($\mathcal{E}$7-$\gamma$6) | 0.064 | 1.83× | 0.064 | 1.91× | 0.074 | 2.16× | 0.070 | 2.09× | 71.93 | 1.99× |
| | FS($\mathcal{E}$7-$\gamma$6) | 0.074 | 2.17× | 0.073 | 2.17× | 0.080 | 2.33× | 0.074 | 2.25× | 80.48 | 2.23× |
| | DV($\mathcal{E}$7) | 0.048 | 1.39× | 0.049 | 1.44× | 0.088 | 2.52× | 0.075 | 2.20× | 67.86 | 1.88× |
| | DEL | 0.087 | **2.41×** | 0.083 | **2.41×** | 0.097 | **2.75×** | 0.084 | **2.44×** | 90.29 | **2.50×** |
| LLaMA -2-13B | Vanilla | 0.025 | 1.00× | 0.025 | 1.00× | 0.025 | 1.00× | 0.025 | 1.00× | 27.08 | 1.00× |
| | LS($\mathcal{E}$7-$\gamma$4) | 0.053 | 1.99× | 0.049 | 1.89× | 0.048 | 1.72× | 0.050 | 1.87× | 50.68 | 1.87× |
| | FS($\mathcal{E}$7-$\gamma$4) | 0.062 | 2.24× | 0.054 | 2.06× | 0.049 | 1.77× | 0.051 | 1.89× | 53.99 | 1.99× |
| | DV($\mathcal{E}$7) | 0.040 | 1.48× | 0.038 | 1.45× | 0.038 | 1.36× | 0.038 | 1.41× | 38.68 | 1.43× |
| | DEL | 0.063 | **2.25×** | 0.056 | **2.11×** | 0.067 | **2.29×** | 0.054 | **1.98×** | 58.39 | **2.16×** |
| LLaMA -2-70B | Vanilla | 0.013 | 1.00× | 0.013 | 1.00× | 0.013 | 1.00× | 0.013 | 1.00× | 9.75 | 1.00× |
| | LS($\mathcal{E}$9-$\gamma$6) | 0.033 | 2.14× | 0.026 | 1.98× | 0.028 | 2.11× | 0.026 | 1.99× | 20.90 | 2.14× |
| | FS($\mathcal{E}$9-$\gamma$6) | 0.036 | 2.76× | 0.027 | 2.09× | 0.029 | 2.22× | 0.026 | 2.00× | 22.08 | 2.26× |
| | DV($\mathcal{E}$9) | 0.021 | 1.59× | 0.019 | 1.50× | 0.020 | 1.51× | 0.019 | 1.51× | 14.86 | 1.52× |
| | DEL | 0.038 | **2.84×** | 0.028 | **2.15×** | 0.035 | **2.57×** | 0.028 | **2.12×** | 23.49 | **2.41×** |

Table 1: Performance comparison of DEL against static and dynamic speculative decoding baselines on text generation tasks. We report empirically observed Token-per-Layer (eTPL), per-task speedup, and overall decoding speed. The best results are bolded and the second-best are underlined. DEL consistently achieves the highest speedups by up to 2.84× across different models and tasks.

(CNN/DM) (Nallapati et al., 2016) for language modeling and summarization, XSUM (Narayan et al., 2018) for abstractive summarization, HumanEval (Chen et al., 2023) for code generation, and AQuA-RAT-CoT (Ling et al., 2017) for arithmetic reasoning.

Following prior work (Elhoushi et al., 2024; Xia et al., 2025; Zhou et al., 2024), we use 1-shot summarization for CNN/DM and 0-shot for XSUM. AQuA-RAT-CoT is evaluated in a 5-shot setting. The maximum generation length is set to 512 tokens. We randomly sample 1,000 test examples per dataset, except for HumanEval and AQuA-RAT, where full test sets are used. For HumanEval, we report both pass@1 and pass@10. Speculative sampling (Leviathan et al., 2023) is used with a batch size of 1. Our implementation builds on the codebase from Elhoushi et al. (2024).

**Baselines. Vanilla** refers to auto-regressive decoding without acceleration. **LS($\mathcal{E}$-$\gamma$)** uses LayerSkip with a fixed exit layer $\mathcal{E}$ and static speculation length $\gamma$. **FS($\mathcal{E}$-$\gamma$)** is a LayerSkip variant with dynamic speculation. It initializes the speculation length to $\gamma$, then adjusts it using a finite-state controller: it increases by 1 if all draft tokens are accepted in the previous round, and decreases by 1 if any are rejected (Liu et al., 2025). **DV($\mathcal{E}$)** is another LayerSkip variant that applies the Draft & Verify method (Zhang et al., 2024), adjusting speculation length dynamically based on a confidence threshold to maintain a target acceptance rate. For all baselines, $\mathcal{E}$ and $\gamma$ are selected via grid search to maximize overall speedup. Detailed experimental setups and results are provided in Appendix B and C.

**Evaluation metrics.** We report the empirically observed Token-per-Layer (eTPL), defined as the total number of generated tokens divided by the number of loaded layers, averaged across all prompts. We also report decoding speed (tokens/s) and wall-time speedup, computed as the relative increase in average inference throughput compared to the auto-regressive baseline under the same settings (Zhang et al., 2024). For output quality, we use ROUGE-2 (Ganesan, 2018), implemented via the TorchMetrics library (Detlefsen et al., 2022)

| Models | CodeLLaMA-7B | | | | | Methods | CodeLLaMA-34B | | | | |
|---|---|---|---|---|---|---|---|---|---|---|---|
| | HumanEval (pass@1) | | HumanEval (pass@10) | | | | HumanEval (pass@1) | | HumanEval (pass@10) | | |
| Methods | eTPL | Speedup | eTPL | Speedup | ROUGE-2 | | eTPL | Speedup | eTPL | Speedup | ROUGE-2 |
| Vanilla | 0.031 | 1.00× | 0.031 | 1.00× | 0.093 | Vanilla | 0.021 | 1.00× | 0.021 | 1.00× | 0.096 |
| LS($\mathcal{E}$7-$\gamma$6) | 0.063 | 1.95× | 0.054 | 1.60× | 0.094 | LS($\mathcal{E}$9-$\gamma$6) | 0.042 | 1.86× | 0.038 | 1.64× | 0.095 |
| FS($\mathcal{E}$7-$\gamma$6) | 0.066 | 2.05× | 0.057 | 1.70× | 0.093 | FS($\mathcal{E}$9-$\gamma$6) | 0.045 | 1.99× | 0.039 | 1.70× | 0.093 |
| DV($\mathcal{E}$7) | 0.069 | 2.08× | 0.048 | $\underline{1.43×}$ | 0.092 | DV($\mathcal{E}$9) | 0.042 | $\underline{1.86×}$ | 0.032 | $\underline{1.43×}$ | 0.099 |
| DEL | 0.077 | **2.26×** | 0.062 | **1.73×** | 0.093 | DEL | 0.051 | **2.12×** | 0.043 | **1.79×** | 0.101 |

Table 2: Comparison of DEL and baselines on code generation tasks using CodeLLaMA-7B and 34B. Pass@1 uses greedy decoding; pass@10 uses sampling with temperature 0.6. We report empirical Token-per-Layer (eTPL), speedup, and ROUGE-2. Best results are bolded; second-best underlined. DEL reliably achieves the highest speedups across both decoding methods. The results indicate consistent output fidelity across methods, including DEL.

## 5.2 Overall Speedup Results

Table 1 shows that DEL consistently outperforms both static (LS) and dynamic (FS, DV) baselines across LLaMA model sizes and text generation tasks. DEL achieves a speedup of up to 2.84× on the AQuA-RAT task with the LLaMA-2-70B model and delivers overall speedups of 2.16× to 2.62× across all models and tasks over vanilla auto-regressive decoding. On LLaMA-2-7B, DEL reaches an overall speedup of 2.50×, exceeding the best baseline (FS) at 2.23× by 0.27× (a 12% speed gain). These gains are driven by higher eTPL, reflecting more efficient layer utilization. DEL maintains strong performance across diverse tasks, including reasoning, summarization, and language modeling, and generalizes well from 1B to 70B models, demonstrating scalability across different tasks and model sizes.

Table 2 reports results on code generation with CodeLLaMA-7B and 34B. DEL achieves the highest speedups under both greedy (pass@1) and sampling-based (pass@10) decoding, reaching up to 2.26× on 7B and 2.12× on 34B. In all cases, it obtains the highest eTPL, confirming its token-level efficiency. For greedy decoding, DEL produces exact token matches with the target model, ensuring identical outputs. In the sampling setting, speculative decoding theoretically preserves the target model's output distribution, which is supported by the ROUGE-2 scores in Table 2. These results confirm that DEL maintains output fidelity, scales effectively with model size, and remains robust across different decoding strategies.

## 5.3 In-depth Analysis

**Ablation study of DEL.** DEL includes two key components: *dynamic exit layer selection* and *dynamic draft-exiting*. To assess their individual contributions, we conduct an ablation study on LLaMA2-7B across all tasks (Figure 3(a)). We evaluate three variants: **DEL-$\mathcal{E}$** disables dynamic draft-exiting and selects both the exit layer and speculation length solely based on the dynamic exit layer selection mechanism (i.e., optimized for TPL). **DEL-$\gamma$** fixes the exit layer and applies dynamic draft-exiting to adapt the speculation length $\gamma$ at runtime. The fixed exit layer in DEL-$\gamma$ is chosen based on the best-performing baseline setting from Table 1. Full DEL, which combines both components, achieves higher speedup than either variant alone, confirming that both contribute to performance gains. **DEL-$\mathcal{E}$-lim** enables both components, but restricts exit layers to $3 < \ell < 32$. The rationale for this study is to explore potential missed opportunities when exit layers are constrained. Even early layers (i.e., layers less than 3) can sometimes yield efficient drafts, and restricting their use reduces potential gains. Thus full flexibility in dynamically selecting exit layers is beneficial.

**Runtime and memory breakdown.** Recall that DEL forwards cached hidden states from all layers through the LM-Head to compute TPL. We quantify the runtime and memory overheads of this additional computation. Figure 3(b) shows the runtime breakdown for different speculation lengths $\gamma$ on LLaMA-2-7B with $\mathcal{E}$=8. The reported runtime is measured after the drafting, verification, and the DEL module has completed processing the corresponding $\gamma$ tokens. In terms of memory, DEL adds only 0.52%~2.26% over vanilla decoding, with minimal overhead compared to LayerSkip's 0.50%~1.42%. These results show DEL is lightweight and easy to integrate. More details are provided in Appendix D.1.

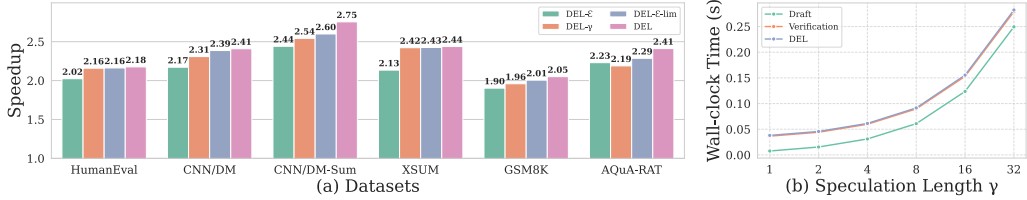

Figure 3: (a) Ablation on LLaMA2-7B shows both dynamic exit layer selection and dynamic draft-exiting components improve performance. (b) Runtime breakdown across $\gamma$ shows DEL adds minimal overhead.

**Evolution of confidence threshold and exit layer.** Figure 4(a) shows how the confidence threshold $\tau$ evolves over 200 SD rounds across different models and datasets. DEL adapts $\tau$ over time, with noticeable variation, showing the importance of using a dynamic rather than fixed threshold. Figure 4(b) visualizes $\tau$ for a sample prompt. The light blue cloud shows the range between confidence scores of matched (top) and mismatched (bottom) shadow tokens. The wide gap makes the midpoint a meaningful threshold for dynamic draft exiting. Figure 4(c) shows how DEL selects the exit layer $\mathcal{E}$ and tracks acceptance rate $\alpha$ during inference. DEL updates $\mathcal{E}$ across rounds to keep $\alpha$ high and maintain efficiency.

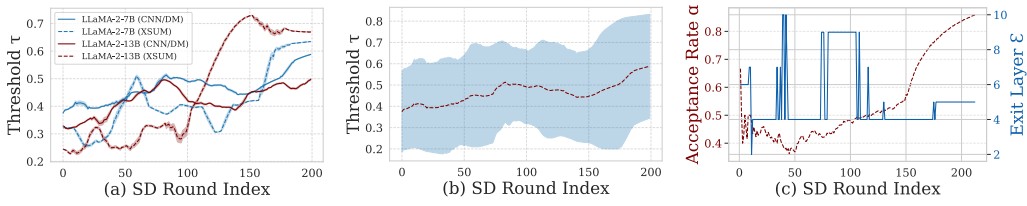

Figure 4: (a) Confidence threshold $\tau$ evolves over 200 SD rounds across models and datasets. (b) $\tau$ for a sample prompt, with accepted (top) and rejected (bottom) token scores shown as a shaded range. (c) Evolution of selected exit layer $\mathcal{E}$ and acceptance rate $\alpha$ during inference.

## 6 Related Works

Early studies on speculative decoding (Leviathan et al., 2023; Chen et al., 2023) introduced a rejection sampling method that preserves the target model's distribution and maintains output quality. Recent work has focused on identifying effective stopping criteria by monitoring confidence scores to trigger verification. Kangaroo (Liu et al., 2024a) halts speculation when the draft model's confidence falls below a fixed threshold, while AdaEDL (Agrawal et al., 2024) estimates a lower bound on token acceptance rate using draft logit entropy. SpecDec++ (Huang et al., 2024) augments the draft model with a trained prediction head to estimate token acceptance rate and dynamically adjust speculation length. CaPE (Du et al., 2024) and SWIFT (Xia et al., 2025) improve speculation accuracy by using confidence scores to expand draft sequences with informative tokens. However, these methods rely on fixed thresholds that do not adapt to varying prompts or tasks. Draft & Verify (Zhang et al., 2024) addresses this limitation by proposing an adaptive, feedback-based thresholding method to enforce a fixed acceptance rate, but it requires extensive hyperparameter tuning.

Self-speculative decoding methods eliminate the overhead of serving two models by using a subset or modified version of the target model to generate draft tokens. Draft & Verify (Zhang et al., 2024), SWIFT (Xia et al., 2025), and Draft on the Fly (Metel et al., 2024) reduce draft generation time by selecting intermediate layers based on Bayesian optimization. Kangaroo (Liu et al., 2024a) uses a shallow sub-network as the draft model, and a lightweight adapter module is trained on it to align its representations with the full model. LayerSkip (Elhoushi et al., 2024) increases early exit accuracy without auxiliary modules through a

specialized training strategy. By performing verification with the remaining layers, it reduces memory usage and benefits from shared computation between drafting and verification. The most relevant approach to ours is S3D (Zhong & Bharadwaj, 2024), which reduces speculation cost by selecting how many mid-layers to skip, but its time-consuming offline training makes it neither plug-and-play nor easily adaptable to different models and tasks. Like prior speculative decoding methods, DEL assumes sufficient alignment between the draft and target distributions; recent works have shown that under out-of-distribution prompts, performance may degrade due to reduced acceptance rates (Hong et al., 2025; Goel et al., 2024; Liu et al., 2024b).

## 7 Conclusion

We introduced DEL, a plug-and-play module for LayerSkip that dynamically selects the exit layer and speculation length to maximize decoding efficiency. DEL tracks token acceptance rates across layers and estimates a Token-per-Layer (TPL) metric, leveraging context-aware feedback to identify the optimal configuration for each speculative decoding round. Extensive evaluations across models and tasks show DEL achieves overall speedups of $2.16\times \sim 2.62\times$ over auto-regressive decoding. DEL's lightweight design enables seamless integration with existing SD pipelines, enhancing performance without sacrificing quality.

## Acknowledgment

We sincerely thank all the reviewers for their time and constructive comments. This material is based upon work supported by NSF award number 2224319, REAL@USC-Meta center, and VMware gift. The views, opinions, and/or findings expressed are those of the author(s) and should not be interpreted as representing the official views or policies of the Department of Defense or the U.S. Government.

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

# A Algorithm

Algorithm 1 summarizes self-speculative decoding with DEL, where $\mathcal{E}$ and $\gamma$ are dynamically adjusted to maximize efficiency and optimize token generation.

---

**Algorithm 1** Self-Speculative Decoding with DEL

---

**Require:**
1:   Target model $\mathcal{M}_p$ with $L$ layers
2:   Initialize hidden states, exit layer $\mathcal{E}$, and threshold $\tau$ by $\mathcal{M}_p$ pre-filling
3:   $r \leftarrow 0$      ▷ Initialize SD round counter
4:   $c, u, tcs, fcs \leftarrow []$      ▷ Lists to store DEL statistics

5: **while** tokens remain to be generated **do**
6:      $\mathcal{D} \leftarrow []$      ▷ Initialize token list
7:      **for** $i = 0$ to $d_{\max} - 1$ **do**      ▷ Draft tokens
8:        Draft token $x_{t+i}$ using the first $\mathcal{E}$ layers with confidence score $s$
9:        **if** $s < \tau$ **then**
10:         **Break**      ▷ Early exit if low confidence
11:       **end if**
12:       Append $x_{t+i}$ to $\mathcal{D}$
13:     **end for**
14:     Verify $\mathcal{D}$ by $\mathcal{M}_p$
15:     DEL_UPDATE($r$)
16:     $r \leftarrow r + 1$
17: **end while**

18: **function** DEL_UPDATE($r$)
19:     $\{(x_t^\ell, cs_t^\ell), \ldots, (x_{t+\gamma}^\ell, cs_{t+\gamma}^\ell)\} \leftarrow$ LM-Head($\{h_t^\ell, \ldots, h_{t+\gamma}^\ell\}$),    $\ell \in [1, L]$
20:     $u_r = \min(\{i \in [0, \gamma] : x_{t+i}^\ell \neq x_{t+i}^L\})$ if mismatch, else $u_r = \gamma$
21:     $c_r^\ell = \sum_{i=0}^{u_r} \mathbb{I}(x_{t+i}^\ell = x_{t+i}^L)$      ▷ Count of matched tokens in valid context
22:     Append $c_r^\ell$ to $c$ and $u_r$ to $u$
23:     $\alpha_\ell = \frac{S(c^\ell)}{S(u)}$      ▷ Calculate acceptance rate
24:     $(\mathcal{E}, \gamma) = \arg\max_{\ell \in [1, L), d \in [0, d_{\max}]} \text{TPL}(\ell, d)$
25:     $tcs_r^\ell = \sum_{i=0}^{u_r} cs_{t+i}^\ell \mathbb{I}(x_{t+i}^\ell = x_{t+i}^L)$      ▷ Compute true confidence scores
26:     $fcs_r^\ell = \sum_{i=0}^{u_r} cs_{t+i}^\ell \mathbb{I}(x_{t+i}^\ell \neq x_{t+i}^L)$      ▷ Compute false confidence scores
27:     Append $tcs_r^\ell$ to $tcs$ and $fcs_r^\ell$ to $fcs$
28:     $\tau_\ell = \frac{1}{2}\left(\frac{S(tcs^\ell)}{S(c^\ell)} + \frac{S(fcs^\ell)}{S(u-c^\ell)}\right)$      ▷ Dynamic threshold update
29:     Set updated exit layer $\mathcal{E}$ and threshold $\tau$ for the next SD round
30: **end function**

---

# B Detailed Experimental Setups

## B.1 Models and Datasets

We evaluate DEL on LLaMA-2, LLaMA-3, and CodeLLaMA models (Touvron et al., 2023; Dubey et al., 2024; Rozière et al., 2024), finetuned for LayerSkip by Elhoushi et al. (2024). Our experiments cover various generation tasks, including language modeling, summarization, arithmetic reasoning, code generation, and translation. For language modeling and summarization, we use CNN/DailyMail (CNN/DM) (Nallapati et al., 2016), while XSUM (Narayan et al., 2018) is used for abstractive summarization. HumanEval (Chen et al., 2023) is used for code generation, and GSM8K (Cobbe et al., 2021) and AQuA-RAT-CoT (Ling et al., 2017) are used for arithmetic reasoning. To cover multilingual generation, we evaluate on the WMT14 De–En dataset (German to English translation). Following prior work (Elhoushi et al., 2024;

Xia et al., 2025; Zhou et al., 2024), we use 1-shot summarization for CNN/DM and 0-shot for XSUM. GSM8K and AQuA-RAT-CoT are evaluated in a 5-shot setting. HumanEval is evaluated on CodeLLaMA-7B and 34B using pass@1 (greedy decoding) and pass@10 (random sampling), with pass@10 results averaged over 10 runs. The maximum generation length is set to 512 tokens. We randomly sample 1,000 test examples per dataset, except for HumanEval and AQuA-RAT, where the full test sets are used. Prompt shots for CNN/DM and GSM8K are randomly sampled from their respective training sets, while AQuA-RAT uses the specific CoT prompt shots provided by Zhou et al. (2024).

## B.2    Inference Setup

To evaluate DEL, we used specific hyperparameter settings. For a given input prompt, DEL initializes $\alpha_\ell$ by pre-filling and considers the last 32 tokens of the context to determine the optimal exit layer $\mathcal{E}$ and speculation length $\gamma$ for the first initial SD round. We apply a decay factor of $\omega = 0.95$ in the weighted sum function in Section 4.2, to gradually forget information from previous SD rounds. For instance, $0.95^{13} \simeq 0.5$ indicates that information from 13 rounds prior is given half the importance of the current round. DEL restricts the maximum speculation length to $d_{\max} = 18$ tokens. For code generation tasks using random sampling, we set the temperature to 0.6 and $top\_p$ to 0.95. All experiments were conducted using PyTorch 2.2.1. Models LLaMA-3.2-1B, LLaMA-2-7B, LLaMA-2-13B, and CodeLLaMA-7B were run on a single NVIDIA A100 (SXM4-40GB) GPU with CUDA 12.8 and 16 CPU cores from an AMD EPYC 7H12 64-Core processor. CodeLLaMA-34B used two instances of the same GPU, while LLaMA-70B ran on two NVIDIA H100 (PCIe-80GB) GPUs with the same CUDA version and 16 CPU cores from an Intel Xeon 6548Y processor. To eliminate hardware variability, we ensured that the same GPU/CPU instances were used consistently for each model-dataset pair. We adopted speculative sampling (Leviathan et al., 2023) as the acceptance strategy with a batch size of 1. Our implementation is based on the codebase from Elhoushi et al. (2024).

## B.3    Baselines

Vanilla refers to standard auto-regressive decoding without acceleration. LS($\mathcal{E}$-$\gamma$) applies LayerSkip with a fixed exit layer $\mathcal{E}$ and a static speculation length $\gamma$. This baseline has two kinds of configurations. LS($\mathcal{E}$-$\gamma$)† uses configurations that are proposed by the original LayerSkip paper, if available. Specifically, ($\mathcal{E}$=8, $\gamma$=6) is applied for HumanEval generation on LLaMA-2-7B, ($\mathcal{E}$=8, $\gamma$=12) for CNN/DM 1-shot and XSUM summarization on LLaMA-2-7B, ($\mathcal{E}$=7, $\gamma$=4) for HumanEval on LLaMA-2-13B, and ($\mathcal{E}$=15, $\gamma$=4) for XSUM summarization. The second configuration kind, LS($\mathcal{E}$-$\gamma$), determines the exit layer and speculation length by conducting a grid search over 10 randomly sampled inputs from a calibration dataset, with a maximum generation length of 256 tokens. Each configuration in the grid search space, consisting of potentially effective combinations of exit layers and speculation lengths for the underlying model, is evaluated on these 10 samples. This process identifies the most effective configurations for the given model and task. FS($\mathcal{E}$-$\gamma$) is a LayerSkip variant with a fixed exit layer $\mathcal{E}$ and dynamic speculation length. It starts with an initial $\gamma$ and adjusts it using a finite-state controller: increasing $\gamma$ by 1 if all draft tokens are accepted in the previous round and decreasing it by 1 if any are rejected (Liu et al., 2025). DV($\mathcal{E}$) is another LayerSkip variant that maintains a fixed exit layer $\mathcal{E}$ but dynamically adjusts the speculation length using the Draft & Verify strategy (Zhang et al., 2024), where the speculation length is adapted based on a confidence threshold to maintain a target acceptance rate.

## B.4    Evaluation Metrics

We use the following metrics to evaluate the performance of DEL. eTPL: empirically observed Token-per-Layer is defined as the total number of generated (verified) tokens divided by the number of loaded layers, averaged across all prompts. For each prompt, we count the total number of loaded layers during generation and compute eTPL by dividing the number of generated tokens by this value. The final reported eTPL is the average across all prompts in the test set. Speed (tokens/s): Speed is defined as the total number of generated

(verified) tokens divided by the total time (seconds) taken to generate the output for a given prompt. This value is computed for each prompt, and the reported speed is the average across all test samples. Speedup: This refers to the ratio of the decoding speed achieved by the given method to that of standard auto-regressive (Vanilla) decoding.

## C Extended Exeprienmental Results

We present the detailed statistics of our main experimental results in Table 3. DEL consistently outperforms both static (LS) and dynamic (FS, DV) baselines across a range of LLaMA model sizes and text generation tasks. On LLaMA-2-70B, DEL achieves a remarkable speedup of 2.84× on AQuA-RAT (Reasoning), outperforming the best FS configuration at 2.78×. For CNN/DM (Abstractive Summarization) on the same model, DEL attains a speedup of 2.57×, while the strongest baseline, FS($\mathcal{E}$9-$\gamma$6), achieves only 2.22×.

| Methods | AQuA-RAT (Reasoning) | | CNN/DM (Lang. Mod.) | | CNN/DM (Abs. Sum.) | | GSM8K (Reasoning) | | HumanEval (Coding) | | XSUM (Abs. Sum.) | | WMT14 De-En (Translation) | |
|---|---|---|---|---|---|---|---|---|---|---|---|---|---|---|
| | Speed | Speedup | Speed | Speedup | Speed | Speedup | Speed | Speedup | Speed | Speedup | Speed | Speedup | Speed | Speedup |
| **LLaMA-3.2-1B** | | | | | | | | | | | | | | |
| Vanilla | 63.17 | 1.00× | 65.15 | 1.00× | 64.23 | 1.00× | 63.91 | 1.00× | 64.84 | 1.00× | 64.83 | 1.00× | 63.92 | 1.00× |
| LS($\mathcal{E}$3-$\gamma$3) | 118.30 | 1.87× | 124.01 | 1.90× | 116.75 | 1.82× | 115.52 | 1.81× | 120.09 | 1.85× | 116.63 | 1.80× | 119.23 | 1.87× |
| LS($\mathcal{E}$3-$\gamma$6) | 127.09 | 2.01× | 137.65 | 2.11× | 124.23 | 1.93× | 117.03 | 1.83× | 126.16 | 1.95× | 122.49 | 1.89× | 127.63 | 2.00× |
| LS($\mathcal{E}$4-$\gamma$3) | 108.31 | 1.71× | 115.48 | 1.77× | 107.13 | 1.67× | 103.97 | 1.63× | 109.44 | 1.69× | 108.26 | 1.67× | 108.74 | 1.70× |
| FS($\mathcal{E}$3-$\gamma$3) | 138.71 | 2.20× | 163.12 | 2.50× | 137.26 | 2.14× | 132.43 | 2.07× | 134.68 | 2.08× | 141.89 | 2.19× | 150.91 | 2.36× |
| FS($\mathcal{E}$3-$\gamma$6) | 136.59 | 2.16× | 161.86 | 2.48× | 137.05 | 2.13× | 130.98 | 2.05× | 135.05 | 2.08× | 141.34 | 2.18× | 151.97 | 2.38× |
| FS($\mathcal{E}$4-$\gamma$3) | 121.20 | 1.92× | 142.35 | 2.18× | 121.66 | 1.89× | 117.15 | 1.83× | 119.74 | 1.85× | 124.56 | 1.92× | 132.81 | 2.08× |
| DV($\mathcal{E}$3) | 108.88 | 1.72× | 125.72 | 1.93× | 117.51 | 1.83× | 90.10 | 1.41× | 107.83 | 1.66× | 92.43 | 1.43× | 94.27 | 1.47× |
| DV($\mathcal{E}$4) | 104.71 | 1.66× | 118.65 | 1.82× | 113.27 | 1.76× | 85.05 | 1.33× | 104.49 | 1.61× | 88.85 | 1.37× | 91.62 | 1.43× |
| DEL | 141.49 | **2.24×** | 167.00 | **2.56×** | 144.78 | **2.25×** | 135.74 | **2.12×** | 140.70 | **2.17×** | 153.63 | **2.37×** | 151.11 | **2.36×** |
| **LLaMA-3-8B** | | | | | | | | | | | | | | |
| Vanilla | 32.85 | 1.00× | 32.57 | 1.00× | 32.68 | 1.00× | 31.90 | 1.00× | 31.65 | 1.00× | 32.16 | 1.00× | 32.29 | 1.00× |
| LS($\mathcal{E}$3-$\gamma$3) | 65.48 | 1.99× | 67.77 | 2.08× | 67.81 | 2.07× | 56.51 | 1.77× | 63.62 | 2.01× | 62.13 | 1.93× | 66.71 | 2.07× |
| LS($\mathcal{E}$3-$\gamma$6) | 71.45 | 2.17× | 74.57 | 2.29× | 75.22 | 2.30× | 54.42 | 1.71× | 71.23 | 2.25× | 66.75 | 2.08× | 74.74 | 2.31× |
| LS($\mathcal{E}$8-$\gamma$6) | 57.11 | 1.74× | 57.90 | 1.78× | 68.54 | 2.10× | 47.50 | 1.49× | 56.91 | 1.80× | 57.61 | 1.79× | 58.57 | 1.81× |
| FS($\mathcal{E}$3-$\gamma$3) | 81.95 | 2.49× | 88.74 | 2.72× | 74.73 | 2.29× | 62.23 | 1.95× | 76.16 | 2.41× | 69.96 | 2.18× | 85.13 | 2.64× |
| FS($\mathcal{E}$3-$\gamma$6) | 81.22 | 2.47× | 89.79 | 2.76× | 75.33 | 2.30× | 61.15 | 1.92× | 75.87 | 2.40× | 70.19 | 2.18× | 84.49 | 2.62× |
| FS($\mathcal{E}$8-$\gamma$6) | 66.51 | 2.02× | 68.36 | 2.10× | 73.89 | 2.26× | 56.34 | 1.77× | 62.58 | 1.98× | 61.86 | 1.92× | 67.85 | 2.10× |
| DV($\mathcal{E}$3) | 49.80 | 1.52× | 49.00 | 1.50× | 75.73 | 2.32× | 44.69 | 1.40× | 47.97 | 1.52× | 47.06 | 1.46× | 49.22 | 1.52× |
| DV($\mathcal{E}$8) | 45.16 | 1.37× | 47.29 | 1.45× | 79.79 | 2.44× | 42.53 | 1.33× | 62.90 | 1.99× | 62.45 | 1.94× | 48.83 | 1.51× |
| DEL | 83.08 | **2.53×** | 91.56 | **2.81×** | 88.97 | **2.72×** | 64.43 | **2.02×** | 78.41 | **2.48×** | 77.21 | **2.40×** | 88.23 | **2.73×** |
| **LLaMA-2-7B** | | | | | | | | | | | | | | |
| Vanilla | 37.08 | 1.00× | 36.05 | 1.00× | 35.61 | 1.00× | 36.31 | 1.00× | 35.98 | 1.00× | 35.59 | 1.00× | 35.37 | 1.00× |
| LS($\mathcal{E}$7-$\gamma$6) | 67.99 | 1.83× | 68.71 | 1.91× | 76.77 | 2.16× | 58.76 | 1.62× | 68.00 | 1.89× | 74.24 | 2.09× | 71.66 | 2.01× |
| LS($\mathcal{E}$8-$\gamma$6)† | 65.65 | 1.77× | 65.94 | 1.83× | 77.96 | 2.19× | 59.83 | 1.65× | 66.81 | 1.86× | 70.55 | 1.98× | 68.27 | 1.92× |
| LS($\mathcal{E}$8-$\gamma$12)† | 57.27 | 1.54× | 58.22 | 1.61× | 78.67 | 2.21× | 48.64 | 1.34× | 59.12 | 1.64× | 67.65 | 1.90× | 62.72 | 1.76× |
| FS($\mathcal{E}$7-$\gamma$6) | 80.50 | 2.17× | 78.40 | 2.17× | 82.98 | 2.33× | 69.68 | 1.92× | 72.27 | 2.01× | 80.02 | 2.25× | 82.25 | 2.31× |
| FS($\mathcal{E}$8-$\gamma$6) | 76.44 | 2.06× | 74.72 | 2.07× | 84.24 | 2.37× | 68.99 | 1.90× | 69.24 | 1.92× | 77.89 | 2.19× | 78.23 | 2.20× |
| FS($\mathcal{E}$8-$\gamma$12) | 74.58 | 2.01× | 72.95 | 2.02× | 82.71 | 2.32× | 66.31 | 1.83× | 66.47 | 1.85× | 75.68 | 2.13× | 74.72 | 2.10× |
| DV($\mathcal{E}$7) | 51.57 | 1.39× | 51.91 | 1.44× | 89.65 | 2.52× | 49.76 | 1.37× | 72.05 | 2.00× | 78.31 | 2.20× | 61.61 | 1.73× |
| DV($\mathcal{E}$8) | 51.27 | 1.38× | 54.57 | 1.51× | 90.78 | 2.55× | 49.26 | 1.36× | 76.06 | 2.11× | 76.91 | 2.16× | 65.61 | 1.84× |
| DEL | 89.43 | **2.41×** | 86.86 | **2.41×** | 98.10 | **2.75×** | 72.31 | **1.99×** | 76.31 | **2.12×** | 86.77 | **2.44×** | 94.42 | **2.65×** |
| **LLaMA-2-13B** | | | | | | | | | | | | | | |
| Vanilla | 27.69 | 1.00× | 27.39 | 1.00× | 25.97 | 1.00× | 27.96 | 1.00× | 27.60 | 1.00× | 27.28 | 1.00× | 27.84 | 1.00× |
| LS($\mathcal{E}$7-$\gamma$3) | 52.95 | 1.91× | 51.48 | 1.88× | 43.81 | 1.69× | 47.27 | 1.69× | 50.01 | 1.81× | 50.20 | 1.84× | 51.20 | 1.84× |
| LS($\mathcal{E}$7-$\gamma$4)† | 55.19 | 1.99× | 51.86 | 1.89× | 44.61 | 1.72× | 47.57 | 1.70× | 50.74 | 1.84× | 51.04 | 1.87× | 52.19 | 1.87× |
| LS($\mathcal{E}$15-$\gamma$4)† | 43.13 | 1.56× | 40.74 | 1.49× | 44.36 | 1.71× | 41.81 | 1.50× | 40.91 | 1.48× | 41.43 | 1.52× | 43.23 | 1.55× |
| FS($\mathcal{E}$7-$\gamma$3) | 63.54 | 2.29× | 57.72 | 2.11× | 45.57 | 1.75× | 52.18 | 1.87× | 52.29 | 1.89× | 52.18 | 1.91× | 54.94 | 1.97× |
| FS($\mathcal{E}$7-$\gamma$4) | 62.04 | 2.24× | 56.48 | 2.06× | 45.94 | 1.77× | 52.37 | 1.87× | 51.96 | 1.88× | 51.51 | 1.89× | 54.87 | 1.97× |
| FS($\mathcal{E}$15-$\gamma$4) | 48.69 | 1.76× | 46.28 | 1.69× | 46.90 | 1.81× | 44.81 | 1.60× | 41.70 | 1.51× | 44.32 | 1.62× | 45.36 | 1.63× |
| DV($\mathcal{E}$7) | 41.03 | 1.48× | 39.77 | 1.45× | 35.39 | 1.36× | 38.72 | 1.38× | 38.99 | 1.41× | 38.53 | 1.41× | 40.75 | 1.46× |
| DV($\mathcal{E}$15) | 40.02 | 1.45× | 36.33 | 1.33× | 50.97 | 1.96× | 37.12 | 1.33× | 47.62 | 1.73× | 48.36 | 1.77× | 46.04 | 1.65× |
| DEL | 62.18 | **2.25×** | 57.89 | **2.11×** | 59.50 | **2.29×** | 52.96 | **1.89×** | 55.51 | **2.01×** | 53.97 | **1.98×** | 55.16 | **1.98×** |
| **LLaMA-2-70B** | | | | | | | | | | | | | | |
| Vanilla | 9.71 | 1.00× | 10.47 | 1.00× | 9.03 | 1.00× | 9.52 | 1.00× | 10.51 | 1.00× | 9.78 | 1.00× | 9.83 | 1.00× |
| LS($\mathcal{E}$8-$\gamma$6) | 24.57 | 2.53× | 20.21 | 1.93× | 18.96 | 2.10× | 17.86 | 1.88× | 20.50 | 1.95× | 19.28 | 1.97× | 19.54 | 2.00× |
| LS($\mathcal{E}$9-$\gamma$3) | 21.55 | 2.22× | 20.02 | 1.91× | 17.92 | 1.99× | 17.69 | 1.86× | 20.07 | 1.91× | 18.90 | 1.93× | 20.55 | 2.10× |
| LS($\mathcal{E}$9-$\gamma$6) | 24.35 | 2.51× | 20.71 | 1.98× | 19.04 | 2.11× | 17.41 | 1.83× | 20.53 | 1.95× | 19.49 | 1.99× | 19.32 | 1.97× |
| FS($\mathcal{E}$8-$\gamma$6) | 26.11 | 2.69× | 20.88 | 1.99× | 19.81 | 2.20× | 17.83 | 1.87× | 20.24 | 1.93× | 18.85 | 1.93× | 19.54 | 2.00× |
| FS($\mathcal{E}$9-$\gamma$3) | 27.00 | 2.78× | 21.79 | 2.08× | 19.87 | 2.20× | 17.88 | 1.88× | 20.47 | 1.95× | 19.31 | 1.97× | 19.98 | 2.04× |
| FS($\mathcal{E}$9-$\gamma$6) | 26.81 | 2.76× | 21.93 | 2.09× | 20.05 | 2.22× | 17.82 | 1.87× | 20.47 | 1.95× | 19.52 | 2.00× | 20.15 | 2.06× |
| DV($\mathcal{E}$8) | 15.37 | 1.58× | 15.57 | 1.49× | 13.66 | 1.51× | 14.17 | 1.49× | 15.65 | 1.49× | 14.66 | 1.50× | 17.52 | 1.79× |
| DV($\mathcal{E}$9) | 15.41 | 1.59× | 15.66 | 1.50× | 13.65 | 1.51× | 14.12 | 1.48× | 15.57 | 1.48× | 14.73 | 1.51× | 17.38 | 1.78× |
| DEL | 27.55 | **2.84×** | 22.48 | **2.15×** | 23.20 | **2.57×** | 19.05 | **2.00×** | 22.90 | **2.18×** | 20.73 | **2.12×** | 20.69 | **2.12×** |

Table 3: Detailed performance comparison of DEL against static (LS) and dynamic (FS, DV) speculative decoding baselines across multiple tasks and model sizes. We report speed (in tokens/sec) and speedup for each method. DEL achieves the highest speedup in most cases, demonstrating consistent gains across different task types and model scales. The best results are bolded, and the second-best results are underlined.

## C.1 Extended Baseline Comparisons

We compare DEL with SWIFT (Xia et al., 2025), a recent self-speculative decoding method, on the HumanEval (code generation) and CNN/DailyMail (language modeling) datasets using LLaMA-2-7B and 13B models. Like DEL, SWIFT is a plug-and-play method that requires no model retraining or architectural modifications. However, unlike DEL, it operates without early-exit capability. To ensure a fair comparison, we evaluate DEL on top of LayerSkip and compare its speedup gain over LayerSkip to the gain SWIFT achieves over vanilla decoding. As shown in Table 4, DEL consistently delivers larger relative improvements across all settings. These results demonstrate that DEL offers stronger acceleration benefits when built atop early-exit self-speculation, highlighting its complementarity with LayerSkip.

| Models | LLaMA-2-7B | | | | | LLaMA-2-13B | | | |
|---|---|---|---|---|---|---|---|---|---|
| | HumanEval | | CNN/DM | | | HumanEval | | CNN/DM | |
| Methods | Speedup | Gain | Speedup | Gain | Methods | Speedup | Gain | Speedup | Gain |
| Vanilla | 1.00× | − | 1.00× | − | Vanilla | 1.00× | − | 1.00× | − |
| SWIFT | 1.01× | **0.01×** | 1.06× | **0.06×** | SWIFT | 1.13× | **0.13×** | 1.26× | **0.26×** |
| LS($\mathcal{E}$7-$\gamma$6) | 1.92× | − | 1.88× | − | LS($\mathcal{E}$9-$\gamma$4) | 1.68× | − | 1.67× | − |
| DEL | 2.07× | **0.15×** | 2.30× | **0.48×** | DEL | 1.85× | **0.17×** | 2.28× | **0.61×** |

Table 4: Speedup comparison of DEL and SWIFT on HumanEval and CNN/DM using LLaMA-2-7B and 13B. "Speedup" is measured relative to vanilla decoding. "Gain" denotes improvement over LayerSkip for DEL, and over Vanilla for SWIFT. DEL consistently achieves larger relative gains than SWIFT across tasks and model sizes.

# D  Further Analysis

## D.1 Memory Breakdown

Table 5 shows the maximum allocated GPU memory during the inference of a randomly selected prompt from each dataset. The experiments were conducted on LLaMA-2-7B using three decoding methods: Vanilla auto-regressive decoding, LayerSkip with exit layer 7 and speculation length 6 (LS($\mathcal{E}$7-$\gamma$6)), and DEL. For each task, the maximum GPU memory used during the entire inference process is reported, along with the memory overhead compared to Vanilla decoding. The results demonstrate that both LS and DEL introduce minimal memory overhead. LS($\mathcal{E}$7-$\gamma$6) incurs an overhead of 0.50%~1.42%, depending on the task, with an average overhead of 0.91% across all tasks. DEL introduces a slightly higher overhead, ranging from 0.52% to 2.26%, with an overall average of 1.54%. Overall, DEL maintains a minimal memory overhead relative to Vanilla decoding, ensuring scalability and making it easy to integrate into various scenarios without significant resource constraints.

| Methods | | HumanEval (Coding) | CNN/DM (Lang. Mod.) | CNN/DM (Abs. Sum.) | XSUM (Abs. Sum.) | GSM8K (Reasoning) | AQuA-RAT (Reasoning) | Overall |
|---|---|---|---|---|---|---|---|---|
| Vanilla | Memory | 13.184 | 13.200 | 14.348 | 13.288 | 14.252 | 14.015 | 13.715 |
| | Overhead | − | − | − | − | − | − | − |
| LS($\mathcal{E}$7-$\gamma$6) | Memory | 13.250 | 13.270 | 14.551 | 13.363 | 14.435 | 14.173 | 13.840 |
| | Overhead | 0.50% | 0.53% | 1.42% | 0.56% | 1.28% | 1.13% | 0.91% |
| DEL | Memory | 13.253 | 13.393 | 14.672 | 13.487 | 14.473 | 14.275 | 13.926 |
| | Overhead | 0.52% | 1.46% | 2.26% | 1.50% | 1.55% | 1.86% | 1.54% |

Table 5: Maximum allocated GPU memory (in GB) during inference of a randomly selected prompt from each dataset using LLaMA-2-7B. Memory overhead relative to Vanilla auto-regressive decoding is shown as a percentage. DEL maintains a minimal memory overhead, ranging from 0.52% to 2.26%, with an overall average of 1.54%, ensuring scalability and ease of integration across different scenarios.

## D.2 Decay Control Factor $\omega$ Sensitivity Analysis

We analyze the impact of different values of $\omega$ on the achieved speedup by running LLaMA-2-7B on the CNN/DM Summarization task. $\omega$ is the decay factor used in the weighted sum function, defined in Section 4.2. This function gradually forgets information from previous SD rounds, and $\omega$ controls the rate of decay. Table 6 presents the results of varying $\omega \in \{0.5, 0.6, 0.7, 0.8, 0.9, 0.95, 1.0\}$. For reference, the first column (with $\omega = -$) shows the baseline results for Vanilla decoding with no speculative acceleration. The table reports eTPL, Speed (tokens/sec), and Speedup for each $\omega$ value. The results indicate that DEL is largely insensitive to $\omega$, with speedup improving from 2.54× at $\omega = 0.5$ to a peak of 2.60× at $\omega = 0.9$ and 1.0, and remaining stable (2.57×) even at $\omega = 0.95$.

| Metrics | $\omega = -$ | $\omega = 0.5$ | $\omega = 0.6$ | $\omega = 0.7$ | $\omega = 0.8$ | $\omega = 0.9$ | $\omega = 0.95$ | $\omega = 1$ |
|---------|------|--------|--------|--------|--------|--------|---------|------|
| eTPL | 0.031 | 0.091 | 0.092 | 0.092 | 0.093 | 0.094 | 0.094 | 0.093 |
| Speed | 35.61 | 90.61 | 91.41 | 91.04 | 92.53 | 92.75 | 91.55 | 92.66 |
| Speedup | 1.00× | 2.54× | 2.57× | 2.56× | 2.60× | 2.60× | 2.57× | 2.60× |

Table 6: Effect of varying $\omega$ on eTPL, speed (tokens/sec), and speedup during inference on the CNN/DM Summarization task using LLaMA-2-7B. DEL remains largely insensitive to $\omega$, achieving near-optimal speedups (up to 2.60×) across a wide range of values.

