# OpenReview forum: "DEL: Context-Aware Dynamic Exit Layer for Efficient Self-Speculative Decoding"
_colmweb.org/COLM/2025/Conference — COLM 2025_

### Official Review · Reviewer_GrNu · 2025-05-11

**Rating:** 6
**Confidence:** 3
**Ethics Flag:** 1

**Summary:**

This paper proposes DEL, a plug-and-play module, in order to help dynamically select the exit layer for drafting ($\epsilon$) and the number of tokens drafted ($\gamma$), two important hyperparameters in speculative decoding. The authors begin with a preliminary study, showing that the optimal choice of exit layer and the speculation length varies in different tasks. Also, the optimal settings vary by prompt and generation stage. Motivated by this, the authors propose DEL, to dynamically select $\epsilon$ and $gamma$ by first presenting a metric TPL as an estimation of the expected number of tokens generated per loaded layer. TPL serves as the criterion for selecting the parameters for the next decoding round. Experiment verified the effectiveness of DEL when plugged into LayerSkip across different models and tasks.

**Reasons To Accept:**

- The proposed method is novel and interesting. The metric TPL could be useful for future works in speculative decoding.
- The experiments are pretty solid on a wide range of tasks and models.
- The paper is a joy to read and well-organized.

**Reasons To Reject:**

- The linked anonymous repository is invalid, which might result in concerns with the reproducibility of this paper.
- I am mainly concerned about the adaptability of this method as a "plug-and-play" module, especially when the experiments are only conducted with LayerSkip. I understand that there might be limited work in the research line of early-exit, but I would be happy to see some discussions on applying this DEL module to other possible methods. Otherwise, the scope of this paper might be limited.
- The experiments are only conducted with the Llama family. Experiments on other model architectures would be beneficial in further strengthening the experimental results.

---

> ### Author Response · Authors · 2025-06-01
>
> We thank the reviewer for the constructive feedback and for highlighting both the strengths and limitations of our work. Below we address your main concerns.
>
> ## Q1: On the anonymous repository link:
> We apologize for the confusion. The hyperlink shown in the OpenReview webpage abstract appears to be misformatted due to translation from latex to markdown, but the link provided in the paper’s abstract is correct and functional.
> The repository can be accessed here without issue:
> https://anonymous.4open.science/r/DEL-F162
>
> ## Q2: On DEL's adaptability beyond LayerSkip
> DEL is designed to operate within early-exit self-speculative frameworks such as LayerSkip, providing dynamic, per-input control over speculation granularity using cached hidden states. As long as the baseline speculative decoding provides these inputs DEL can be adapted.
>
> Early-exit self-speculation schemes like LayerSkip offer an efficient foundation by eliminating the need for a separate draft model, avoiding additional memory for KV cache or model parameters, and preventing recomputation of draft layers during verification. As early-exit capabilities become increasingly common, we hope DEL provides an interesting solution that other researchers may build on and adapt to such systems.
>
> ## Q3: On evaluation beyond LLaMA family models
> We focused on the LLaMA family due to the availability of early-exit (LayerSkip-tuned) checkpoints. However, DEL is model-agnostic and can be applied to any decoder-style transformer that supports early exits and exposes intermediate activations, as we demonstrated in the additional experiments on the latest Llama-3-8B model (see responses for reviewer #SrG8). We anticipate that more out-of-the-box LayerSkip-tuned models beyond the LLaMA family will become available, all of which can be directly used with DEL.

---

> > ### Author Response · Authors · 2025-06-09
> >
> > Thank you again for your thoughtful and encouraging review.
> >
> > In our response, we clarified the correct anonymous repository link, explained how DEL can be adapted to other early-exit frameworks beyond LayerSkip, and noted that DEL is model-agnostic, with added results on LLaMA-3-8B to demonstrate broader applicability.
> >
> > With less than two days left in the discussion, and each review being impactful, we'd be grateful for any further feedback. If our responses addressed your concerns, we would appreciate it if you could consider updating your review and score.

---

### Official Review · Reviewer_SrG8 · 2025-05-13

**Rating:** 7
**Confidence:** 3
**Ethics Flag:** 1

**Summary:**

This paper proposes a novel approach to adaptively select the exit layer in LayerSkip self-speculative decoding. The core idea is that number of layers required for a good draft prediction depends on the context so far, sometimes it has to be more or less depending on how complicated so far context was. Authors propose a way of how to estimate such complexity or trade off between speed and acceptance rate. The proposed DEL algorithm is using so called shadow tokens and cached layer activation to plan which exit layer should be used for a given prompt context. Experimental results confirm that the proposed approach optimizes such tradeoff very effectively that results in practical speed ups in different target tasks.

**Questions To Authors:**

* Do you know if self-speculative decoding can be applied to LoRA adapter trained models? I.e. if early exit work as well when we use lower rank approximation parameters during decoding?

**Reasons To Accept:**

* Self speculation is very practical algorithm, and this DEL adaptive version has all chances to be widely used in the community, so its a good contribution.

**Reasons To Reject:**

* Model selection is quite outdated given LLama2 and CodeLLama models. From my understanding that choice aligns well with baseline results from LayerSkip methods, but i will be happy to see how this method confirms its efficiency with newer models such as qwen3 and llama3.

---

> ### Author Response · Authors · 2025-06-01
>
> We thank the reviewer for the thoughtful feedback and for recognizing DEL’s practicality and potential impact. Below we address the raised concerns and questions.
>
> ## Q1: On the model selection being limited to LLaMA-2 and CodeLLaMA
> We focused on LLaMA-2 and CodeLLaMA due to the availability of LayerSkip-tuned versions. That said, DEL is model-agnostic and we believe can be adapted to any decoder-style transformer that supports early exits and exposes intermediate activations.
>
> To demonstrate DEL’s applicability to newer models, we now include additional results using LayerSkip-tuned LLaMA-3-8B, expanding beyond the previously reported LLaMA-3.2-1B results in the main paper. These experiments confirm that DEL continues to deliver consistent improvements across tasks, even on latest models:
>
> | Method    | CNN/DM (Abs) | XSUM      | CNN/DM (LM) | HumanEval | GSM8K     | Overall   |
> | --------- | ------------ | --------- | ----------- | --------- | --------- | --------- |
> | Vanilla   | 1.00X        | 1.00X     | 1.00X       | 1.00X     | 1.00X     | 1.00X     |
> | LS(E3-γ3) | 2.00X        | 1.90X     | 1.98X       | 1.94X     | 1.73X     | 1.91X     |
> | FS(E3-γ3) | 2.19X        | 2.11X     | 2.56X       | 2.29X     | 1.86X     | 2.20X     |
> | DV(E3)    | 2.21X        | 1.45X     | 1.46X       | 1.47X     | 1.40X     | 1.60X     |
> | DEL       | **2.58X**    | **2.32X** | **2.61X**   | **2.37X** | **1.93X** | **2.36X** |
>
>
> ## Q2: On compatibility with LoRA adapter-based models
> DEL does not impose any assumptions on the underlying model architecture beyond early-exit support and access to intermediate hidden states. It is fully compatible with models fine-tuned via parameter-efficient methods such as LoRA.
>
> In fact, early-exit capabilities can be incorporated during fine-tuning using LoRA adapters, and LayerSkip training itself can be adapted for PEFT methods. As these training approaches become more common, we expect DEL to be widely deployable, serving as a general plug-and-play controller for a range of self-speculative frameworks regardless of whether the base model uses full fine-tuning, adapters, or other lightweight tuning strategies.

---

> > ### Comment · Reviewer_SrG8 · 2025-06-08
> >
> > thanks for adding new results! Looking forward to seeing it at the conference.

---

> > > ### Author Response · Authors · 2025-06-09
> > >
> > > Thank you for your response, we appreciate it. We welcome any further comments or discussion.

---

### Official Review · Reviewer_TsAY · 2025-05-13

**Rating:** 6
**Confidence:** 4
**Ethics Flag:** 1

**Summary:**

This paper proposes DEL (Dynamic Exit Layer), a light‑weight, plug‑and‑play controller that augments LayerSkip self‑speculative decoding with fine‑grained, context‑aware decisions about where to exit the network (number of transformer layers E) and how long to speculate (token block size γ).  DEL monitors layer‑wise acceptance rates in real time by re‑routing cached hidden states through the LM head to create “shadow” predictions, from which it derives a Token‑per‑Layer (TPL) efficiency metric and an adaptive confidence threshold for early stopping.  Across six generation tasks, four LLaMA sizes (1 B–70 B) and CodeLLaMA (7 B/34 B), DEL delivers 2.16×–2.50× overall speed‑ups over vanilla decoding and up to 0.27× over the best existing dynamic speculative baselines while maintaining output quality, all with <2.3 % extra memory and negligible latency overhead.  The manuscript is clearly written, provides intuitive motivation, thorough algorithmic detail, and ablation studies that isolate the benefit of dynamic E and dynamic γ individually.

Conceptually, the work is an incremental yet meaningful advance: it shows that speculative‑decoding hyper‑parameters are highly instance‑dependent and offers a practical solution that requires no retraining or extra models.  While the ideas build on LayerSkip and existing confidence‑based heuristics, the unified, on‑the‑fly optimisation of both E and γ is original, and the empirical section is broad enough to suggest significance for both research prototypes and production LLM inference stacks.

**Reasons To Accept:**

- DEL is a drop‑in module that piggy‑backs on hidden states already computed by LayerSkip, adding <2 % memory and minimal time overhead, which lowers the barrier to deployment in latency‑sensitive settings.
- On every model‑task pair tested—spanning reasoning, summarisation, language modelling and code generation—DEL secures the best or second‑best speed‑ups, with empirical Token‑per‑Layer gains confirming that efficiency, not luck, drives the results (Tables 1–2).
- The TPL metric and decayed statistics for acceptance‑rate estimation are simple, interpretable, and validated via ablations that show each component’s contribution.
- Experiments cover models from 1 B to 70 B parameters and both greedy and sampling decoding, demonstrating robustness across sizes and decoding regimes.
- The paper provides pseudo‑code, hyper‑parameter details, and an open‑source link, facilitating replication and extension.

**Reasons To Reject:**

- DEL implicitly assumes access to a LayerSkip‑tuned model; it is unclear how well the controller would work with off‑the‑shelf checkpoints or other self‑speculative variants.
- Apart from pass@k on HumanEval and reporting that output distributions are “unchanged,” the paper offers no human or automatic fidelity metrics (e.g., BLEU, ROUGE, MMLU), leaving open the risk of subtle quality degradation.
- The TPL cost model relies on a memory‑bound assumption and a fixed decay factor ω=0.95; sensitivity analyses are absent, and no theoretical guarantee links maximising TPL to minimising latency.
- All tasks are English and transformer‑decoder based; effects on multilingual prompts, encoder–decoder models, or dialogue safety filters are unexplored.
- The controller could oscillate or choose poor configurations on out‑of‑distribution prompts, but stability under such conditions is not studied.

---

> ### Author Response · Authors · 2025-06-01
>
> Thank you for your detailed and thoughtful review. We are grateful for your recognition of DEL's practical significance and empirical strength. Below, we address each of your concerns and suggestions.
>
> ## Q1: On reliance on LayerSkip-tuned models
> DEL is designed to operate within early-exit self-speculative frameworks such as LayerSkip, providing dynamic, per-input control over speculation granularity using cached hidden states. As long as the baseline speculative decoding provides these inputs DEL can be adapted.
>
> Early-exit self-speculation schemes like LayerSkip offer an efficient foundation by eliminating the need for a separate draft model, avoiding additional memory for KV cache or model parameters, and preventing recomputation of draft layers during verification. As early-exit capabilities become increasingly common, we hope DEL provides an interesting solution that other researchers may build on and adapt to such systems.
>
> ## Q2: On missing fidelity metrics
> For greedy decoding scenarios, we perform exact match to accept a speculated token by the target model. As a result, the generated outputs are identical, and thus fidelity is always preserved.
>
> For sampling-based decoding, we  provide ROUGE-2 fidelity results for DEL on HumanEval using CodeLLaMA-7B. Results are averaged across 10 runs:
>
> | Method     | ROUGE-2 |
> |------------|---------|
> | Vanilla    | 0.09350 |
> | LS(E7-γ6)  | 0.09343 |
> | FS(E7-γ6)  | 0.09283 |
> | DV(E7)     | 0.09261 |
> | DEL        | 0.09283 |
>
> The results indicate consistent output fidelity across methods, including DEL. We will include these results and clarifications in the final revision.
>
> ## Q3:
> *On the TPL cost model and its memory-bound assumption.* Section 5.3 (Figure 3b) provides a runtime breakdown showing that verification latency remains nearly flat as the number of verified tokens increases (1 to 32 tokens), confirming the memory-bound behavior. This justifies TPL as a suitable proxy for decoding cost and latency.
>
> *On the ablation of $\omega$.* As reported in Appendix D.2 (Table 5), DEL is robust to the decay factor $\omega$. Speedup is stable in range 2.54X to 2.60X for different $\omega$ in range [0.5, 1]. This stability supports ease of DEL's deployment without requiring hyperparameter tuning.
>
> *Theoretical guarantee of TPL.* While we do not present a formal derivation, Section 4.1 defines TPL as a speculation utility metric under the assumption that latency is dominated by the memory bandwidth of the underlying hardware, a common assumption among Speculation Decoding works, that is empirically validated for our setting in Section 5.3 (Figure 3b). Furthermore, the effectiveness of TPL is empirically verified by the results reported in Section 5.2 (Table 1) indicating the strong correlation between TPL improvements and actual wall-clock speedups observed across models and tasks.
>
> ## Q4: On coverage of model architectures and language settings
> DEL currently targets decoder-only models, which are common in generative tasks. However, the formulation is architecture-agnostic and can extend to encoder-decoder models with early-exit support.
>
> To explore multilingual applicability, we ran WMT14-DE-EN translation with LLaMA-2-7B. Results:
>
> | Method     | eTPL  | Speed (tok/s)  | Speedup |
> |------------|-------|----------------|---------|
> | Vanilla    | 0.031 | 53.38          | 1.00X   |
> | LS(E7-γ6)  | 0.069 | 106.08         | 1.99X   |
> | FS(E7-γ6)  | 0.080 | 123.08         | 2.31X   |
> | DV(E7)     | 0.059 | 90.33          | 1.69X   |
> | DEL        | 0.096 | 136.57         | **2.56X** |
>
> DEL achieves a 2.56X speedup over vanilla, exceeding the best baseline (2.31X). The full set of experiments with the multilingual task will be included in the final revision.
>
> ## Q5: On controller stability under out-of-distribution prompts
> DEL does not assume in-distribution prompts. The controller makes decisions based on observed acceptance rates and internal confidence signals at inference time. However, like other speculative decoding methods, DEL works best when an alignment exist between draft and target model distributions, which is generally needed for any speclative decoding to work reasonably. Under significant distribution shift, draft errors may increase, reducing effective speculation length.
>
> OOD robustness is an orthogonal direction for Speculative Decoding domain. We will expand our discussion of this point in Section 6 of the paper.

---

> > ### Author Response · Authors · 2025-06-09
> >
> > Thank you again for your thoughtful review.
> >
> > In our response, we clarified DEL's adaptability beyond LayerSkip-style setups, added ROUGE-2 results showing DEL preserves fidelity under sampling, provided empirical support for the TPL model and its stability across $\omega$ values, included multilingual results with notes on encoder–decoder applicability, and clarified DEL's adaptivity under OOD prompts, with plans to expand this discussion in the Related Works Section.
> >
> > With less than two days left in the discussion, and each review being impactful, we'd greatly appreciate any further feedback. If our responses have addressed your concerns, we'd be grateful if you could consider updating your review and score.

---

### Official Review · Reviewer_wiH3 · 2025-05-19

**Rating:** 6
**Confidence:** 4
**Ethics Flag:** 1

**Summary:**

This paper aims to improve a subclass of speculative decoding methods that employ an early exit (sub)model of the target/verifier LLM as the draft model. The paper first conducts an empirical study to assess the effect of the choice of the early exit layer $\mathcal{E}$ and speculation length $\gamma$ -- the number of tokens produced by the draft model before invoking the verification phase. The empirical study shows that the optimal choice of $(\mathcal{E}, \gamma)$ is model- and input-dependent. Motivated by this, the paper proposes an adaptive approach, namely DEL, to select $(\mathcal{E}, \gamma)$. Towards this, the paper introduces a metric called token-per-layer (TPL) that characterizes the utility of selecting a particular exit layer to define the draft model. The paper proposes an efficient approach to estimate the expected acceptance rate for the tokens drafted by each layer, which in turn defines the TPL metric. Subsequently, DEL selects the early exit layer by optimizing the TPL metric. As for the dynamic selection of  $\gamma$, the paper compares the prediction confidence with a dynamic threshold to decide if the speculation should be stopped and the verification phase needs to be invoked. The paper evaluates the utility of DEL on 4 benchmarks and shows that it outperforms various baselines. The paper also performs an ablation study to showcase the value of the dynamic selection of both $\mathcal{E}$ and $\gamma$.

**Questions To Authors:**

- How is the TPL metric different from the improvement factor considered by Leviathan et al. (https://arxiv.org/pdf/2211.17192; Theorem 3.8) in a meaningful way?
- Could the authors elaborate on "Through extensive evaluations across tasks and model sizes,...and **0.27× improvement** over dynamic LayerSkip baselines"?

**Reasons To Accept:**

- The paper conducts a systematic study of the effect of $(\mathcal{E}, \gamma)$ on early exit-based drafting strategies for speculative decoding.
- The paper proposes DEL -- an approach to dynamically select $(\mathcal{E}, \gamma)$  for speculative decoding.
- The paper presents empirical results showcasing the utility of DEL on multiple benchmarks. In addition, the ablation results clearly establish the importance of dynamically adjusting both $\mathcal{E}$ and $\gamma$.

**Reasons To Reject:**

- Given that multiple early exit-based speculative decoding methods already present in the literature, covering the selection of the early exit layer $\mathcal{E}$ and utilization of model confidence-based speculation lengths, the novelty of the contributions is somewhat limited.
- The gains realized by DEL are not significantly large compared to other baselines considered in the paper.
- The empirical evaluation in the paper can be expanded to cover more baselines, such as S3D and Eagle.

---

> ### Author Response · Authors · 2025-06-01
> **Official Comment by Authors (1/2)**
>
> We thank the reviewer for the thoughtful and detailed feedback. We're encouraged that you found the empirical study and dynamic control of early-exit based self-speculative decoding valuable. Below, we address your comments and questions.
>
> ## Q1: On the Novelty of DEL
> DEL is, to our knowledge, the first controller to jointly and dynamically select both the exit layer ($\mathcal{E}$) and speculation length ($\gamma$) at runtime, per input, without any retraining, architecture modifications, or offline hyperparameter finetuning.
>
> What sets DEL apart is its context-based, plug-and-play design. It leverages cached hidden states already produced during LayerSkip-style inference to evaluate draft utility in real-time with negligible overhead. This allows DEL to adaptively optimize the tradeoff between latency and acceptance rate using a unified control policy, rather than static or separately tuned heuristics for $\mathcal{E}$ and $\gamma$.
>
> We believe this unified, context-aware formulation allows effective dynamic speculative decoding, particularly in practical deployment settings where per-input efficiency is critical.
>
> ## Q2: On the performance gains
> Recent infrastructure studies [1, 2] demonstrated that performance optimizations in AI inference, even if modest, have high impact in industrial settings.  DEL improves the decoding speed from 2.23X (FS baseline) to 2.5X over no speculation decoding baseline.  DEL achieves these speedups over SOTA baseline (FS) and across a wide range of models (1B–70B) and tasks (reasoning, summarization, language modeling, code generation), all with zero offline overhead and no need for model modifications.
>
> These gains are especially compelling given that it requires no retraining, no hyperparameter tuning, and integrates seamlessly into early-exit self-speculation systems, like LayerSkip.  We respectfully argue that ease of adoption and consistent gains are valuable in practical deployment scenarios.
>
> [1] https://ai.google.dev/edge/mediapipe/solutions/genai/llm_inference
>
> [2] https://cloud.google.com/blog/products/compute/accelerating-ai-inference-with-google-cloud-tpus-and-gpus
>
> ## Q3: On expanding baselines
> Based on the request for expanding baselines, during the rebuttal window we compared DEL with another SOTA baseline SWIFT (Xia et al., ICLR 2025), a recent self-speculation method that, like DEL, requires no retraining or architectural modification. However, SWIFT operates without early-exit capability.  Below are results comparing DEL to SWIFT and LayerSkip on LLaMA-2-13B:
>
> | Method    | HumanEval (Speedup) | CNN/DM (Speedup)   |
> | --------- | ------------------- | ------------------ |
> | Vanilla   | 1.00X               | 1.00X              |
> | SWIFT     | 1.13X (**+0.13X**)  | 1.26X (**+0.26X**) |
> | LayerSkip | 1.68X               | 1.67X              |
> | DEL       | 1.85X (**+0.17X**)  | 2.28X (**+0.61X**) |
>
> These results show that DEL achieves stronger gains atop LayerSkip than SWIFT does over vanilla decoding. We will include the full set of experiments and clarifications in the final revision.
>
> EAGLE and S3D make training adaptations to the model architecture and token search process for improving drafting, which is an orthogonal approach to DEL. For example, EAGLE performs tree-structured decoding to increase acceptance rate, and train specific head atop the target model to generate drafts.  DEL on the other hand is a plug-and-play controller that dynamically adjusts the exit layer and speculation length without requiring any changes to the training process.

---

> > ### Author Response · Authors · 2025-06-01
> > **Official Comment by Authors (2/2)**
> >
> > ## Q4: On TPL vs. Leviathan et al.’s Improvement Factor
> > Leviathan et al. derive a theoretical speedup factor for speculative decoding based on the expected acceptance rate $\alpha$ and a fixed speculative length $\gamma$, assuming i.i.d. acceptance events. Their analysis models cost in terms of the number of speculative tokens generated before fallback verification with the main model, focusing on token-level improvement and the tradeoff between draft length and acceptance probability.
> >
> > DEL adapts this idea in the context of LayerSkip, where decoding cost is primarily determined by the number of transformer layers invoked during draft and verification, rather than just the number of speculative tokens. Specifically, DEL evaluates speculative cost based on both the exit layer $\mathcal{E}$ (how deep the draft computation goes) and the speculation length $\gamma$ (how many tokens are generated speculatively), both of which are chosen dynamically at inference time.
> >
> > Moreover, instead of assuming a fixed $\alpha$, DEL estimates token-level acceptance likelihoods on-the-fly using shadow tokens generated from cached hidden states. This allows DEL to dynamically adjust speculative granularity (both in depth and length) on a per-input basis, enabling adaptive control over the latency-accuracy tradeoff.
> >
> > ## Q5: On the 0.27X Improvement Statement
> > Sorry for the confusion. This data was refering to the results reported in Section 5 (Table 1). For LLaMA-2-7B, the best dynamic baseline (FS), which supports variable speculation length, achieves a 2.23X speedup over vanilla decoding. DEL improves this speedup to 2.50X.

---

> > > ### Comment · Reviewer_wiH3 · 2025-06-08
> > >
> > > Thank you for providing additional experiments, including SWIFT, and answering my questions. I have slightly increased my score. That said, my concerns about the limited novelty remain.

---

> > > > ### Author Response · Authors · 2025-06-09
> > > >
> > > > Thank you for revisiting our response and for your thoughtful engagement with our work. We appreciate your acknowledgment of the new results and your updated assessment. Regarding novelty, we believe we have addressed this aspect to the best of our ability by clarifying DEL's unified, plug-and-play approach and its distinction from prior work. If there are any specific points you feel remain unaddressed, we would be glad to discuss and clarify them further in the remaining discussion period.

---

### Decision · Program_Chairs · 2025-07-08

**Decision:**

Accept

**Comment:**

Speculative decoding has become an active area of work and this paper proposes some innovations in that direction. Reviewers generally consider that the proposed approach for determining an exit layer is an original contribution and has a chance to be widely adopted even in combination of other speculative decoding methods. There are some reservations with respect to similar competing approaches and hence the contribution being incremental however all reviewers seem to acknowledge still some contribution and weigh mostly on the positive side. Looking at the overall picture the AC is also inclined to recommend acceptance.